# High-Embedded Low-Distortion Multihistogram Shift Video Reversible Data Hiding Based on DCT Coefficient

**Yuhang Yang, Xuyu Xiang \*, Jiaohua Qin, Yun Tan, Zhangdong Wang and Yajie Liu**

Department of Computer and Information Engineering, Central South University of Forestry and Technology, Changsha 410004, China
* Correspondence: xyuxiang@csuft.edu.cn

**Abstract:** Video reversible data hiding technology can be applied to copyright protection, medical images, the military, and other fields, but it cannot guarantee high visual quality with an effective embedded capacity. In this paper, a high-embedding and low-distortion reversible data hiding scheme based on a discrete cosine transform (DCT) coefficients method is proposed. The scheme first decodes the original video stream with entropy, obtains all the DCT blocks, and selects the embeddable DCT blocks according to the capacity of the zero factor. Then, it divides the coefficients in the DCT blocks into the shift and embedding coefficients. The shift coefficients directly generate a one-dimensional histogram; the embedding coefficients generate a two-dimensional histogram according to paired strategies. Finally, the secret data can be successfully embedded according to the proposed two-dimensional histogram shift reversible data hiding scheme. This scheme performed more effectively than existing schemes in terms of the embedded capacity, peak signal-to-noise ratio (PSNR), and structural similarity index measure (SSIM).

**Keywords:** reversible data hiding; histogram shift; H.264/AVC video; multiple histograms; computer vision

## 1. Introduction

Now, with the promulgation of a series of laws and regulations to protect original works from infringement, one way to prevent copyright infringement is to hide the copyright information in the video. Tan et al. [1] proposed this method through their robust non-blind watermarking schemes in YCbCr color space based on channel coding. The peak signal-to-noise ratio of the watermarked image and the normalized correlation coefficient of the extracted watermark are investigated. However, existing watermarking technology damages the original content. Agilandeeswari et al. [2–5] proposed four video watermarking schemes from 2015 to 2018; their proposed method has promising prospects in terms of robustness, impeccability, and security, which can be applied to most famous image processing attachments, geometric attachments, video processing attachments, and variable traditional watermarking algorithms. Luo et al. [6] proposed a coverless image steganography method based on multi-object recognition. This method has better performance in terms of robustness and resisting most image attacks. However, steganography technology is irreversible. Reversible data hiding can embed and extract data without damaging the original data. Therefore, by using reversible data hiding technology for copyright protection, one can achieve the expected results. Additionally, reversible data hiding technology applications include, but are not limited to, copyright protection, such as in military applications, medical imaging, and so on. Videos are indispensable to everyone today, and thus the reversible data hiding technology of videos urgently needs to be enhanced. Therefore, we conducted an in-depth study on the reversible data hiding of videos.

People can implement reversible data hiding in three ways: integer transformation [7–9], histogram shift (HS), or differential expansion (DE). Most reversible data hiding

schemes cannot be directly applied to videos, but videos are more common in everyday life than images. Based on this, Chung et al. [10] proposed a histogram shift method to embed motion vectors (MVs) in DCT coefficients for H.264/AVC videos, which implements intraframe error hiding. Fallahpour et al. [11] embedded data in the last nonzero coefficient of a DCT block, which reduced the hidden distortion but was inefficient to embed. Niu et al. [12] added to [10] by proposing an MVs method based on a two-dimensional histogram shift, which further enhanced the embedding efficiency. Tian et al. [13] proposed a new reversible data embedding method for digital images. They explored the redundancy in digital images to achieve a high embedding capacity or to maintain low distortion. However, these methods cannot achieve both. Therefore, this paper proposes a new two-dimensional histogram shift scheme to solve the embedding distortion problem by using a multihistogram shift, which can maintain high embedding while having a low embedding distortion.

The histogram shift principle shifts all the storage units between the peak storage unit and the zero-point storage unit towards zero. Embedding secret information by moving the histogram storage unit requires additional information to extract or restore the original data, which exists as peaks and zeros. Based on the limitations of existing histogram technology in terms of both embedded capacity and video quality performance, this paper proposes a new histogram shift scheme. First, the scheme transcodes the YUV file to obtain an H.264 video stream, and then it applies entropy decoding to obtain a DCT block. Then, using the zero-factor capacity, it selects the DCT block that can be embedded in the data; it obtains the DCT coefficient. Secondly, the scheme divides the coefficients into the shift and embedding coefficients to generate one- and two-dimensional histograms, respectively. Then, the new two-dimensional histogram shift scheme that we propose embeds the data, which can result in data with high efficiency, capacity, and visual quality. The experimental results showed that the scheme is effective. Compared with the related schemes, our scheme had a higher embedding capacity, more efficient performance, and higher visual quality in the peak signal-to-noise ratio and structural similarity index measure.

To use the histogram shift scheme more efficiently, this paper proposes and studies a multihistogram shift scheme based on DCT coefficients. The contributions of this work are summarized as follows:

(1)　We propose a new construction scheme to generate multiple histograms that solves the video quality degradation problem.
(2)　We present a two-dimensional histogram shift strategy based on DCT coefficients that solves the low embedding problem of existing schemes.
(3)　This scheme achieves a more efficient performance in the embedded capacity, peak signal-to-noise ratio, and structural similarity index measure.

In Section 2, we briefly review some of the relevant preparations for this article. In Section 3, we present a multihistogram shift scheme based on DCT coefficients. In Section 4, we present the experimental results based on the scheme. Finally, in Section 5, we draw a conclusion.

## 2. Related Works

Scholars implement today's data embedding algorithms in either the spatial or frequency domains, which have different characteristics. Therefore, they are not comparable, but they can learn from each other. The main reversible data hiding methods that researchers use are the differential expansion (DE) method proposed by Tian et al. [13] in 2003 and the histogram shift (HS) method proposed by Ni et al. [14] in 2006. The differential expansion method is developed by the difference between two consecutive pixel values. First, the difference is made up of the difference between adjacent pixel values. Then, the difference is expanded, and the secret data is embedded in the difference. Finally, the extended difference is used to obtain the corresponding hidden values, and these values cannot be less than 0 or greater than 255, which is called the underflow and overflow, respectively. This method can cause considerable distortion. Lou et al. [15] proposed reduc-

ing the difference before embedding to minimize noise. Subsequently, Ou et al. [16] and Liu et al. [17] further explored differential expansion by proposing a prediction error extension and variation, which helps to increase the data quality. Liu et al. [18] proposed a novel coverless image steganography algorithm based on image retrieval of DenseNet features and DWT sequence mapping. A DenseNet convolutional neural network model in deep learning is used to extract features from image datasets. This method has better robustness and security performance, resisting most image attacks. Ren et al. [19] proposed an RDH-EI scheme based on an adaptive prediction-error label map. Extensive experimental results have shown that the proposed method can achieve effective pixel prediction results and obtain a higher embedding rate (ER).

In the histogram shift method, the histogram is generated from the values in the frame or image. The idea of embedding is to move the histogram to provide space for the peaks, which can be embedded. In this algorithm, the histogram only moves toward zero, and the number of embeddings corresponds to the number of bits at the peak. On this basis, Wu et al. [20] presented a new 2D histogram-based contrast enhancement with reversible data hiding (CE-RDH) scheme that takes brightness preservation (BP) into account. Experimental results on three color image sets demonstrate the efficacy and reversibility of the proposed scheme. Chang et al. [21] proposed a color image RDH method based on adaptive map selection. This scheme is used to solve the problem that the reversible mapping design for color images in the RDH method ignores specific image content, resulting in limited embedding performance. Zhang et al. [22] proposed the adaptive modification for multiple prediction-error histograms (PEHs). An iterative self-learning optimization algorithm is devised to adaptively generate the 2D mapping based on the PEH and payload. A loss function is employed, and the optimization can be solved linearly to reduce the time cost.

Now that video is a popular format, people can naturally apply reversible data hiding to video content, especially in the H.264/AVC [23] and HEVC formats. Ning et al. [24] aimed to solve the problem that existing motion vector data hiding algorithms based on two-dimensional histogram modification do not fully utilize the embedding space; a large-capacity video reversible data hiding algorithm is proposed. This scheme combines the idea of two-layer embedding to further increase the embedding capacity. Chen et al. [25] proposed a novel separable scheme for encryption and reversible data hiding. The embedding algorithm used in the proposed scheme can provide a higher capacity in the video with a lower quantization parameter and a good visual quality of the labeled decrypted video, maintaining low bit rate variation. The video encryption and reversible data hiding are separable, and the scheme can be applied in more scenarios. Kim et al. [26] proposed hiding secret information in the midfrequency region, which hides four bits of secret information in each DCT block without increasing or distorting the file size due to embedding data. Chen et al. [27] made the last change in the QDCT coefficient to minimize the visual quality. They embedded the three bits of information into a pair of coefficients, which had a high peak signal-to-noise ratio with a considerable increase in the embedded capacity while maintaining a small payload.

Recently, Chen et al. [28] proposed a two-dimensional reversible data hiding scheme with a high embedding capacity based on [27]. The scheme achieved a higher embedded capacity without too much degradation in the peak signal-to-noise ratio and similarity. Chen et al. [29] also proposed a scheme that combines histogram correction with run-level encoding in H.264/AVC. The scheme enhances the bit rate while reducing SSIM changes. Ahmad et al. [30] addressed the embedded capacity and file quality issues by studying the histogram of frame formation in video files and the prediction error between two consecutive frames. Yao et al. [31] implemented an effective reversible data hiding scheme in the encrypted video stream and theoretically analyzed the distortion problem during embedding and subsequent interframe distortion drift. Xu et al. [32] further optimized the embedding strategy with the two-dimensional histogram shift method and proposed two new two-dimensional histogram shift schemes, one using DCT coefficient

pairs and the other not using DCT coefficient pairs. In most cases, this scheme achieved high visual quality without considerably increasing the file size. Kang et al. [33] used a multidimensional histogram shift scheme for the QDCT coefficients in the H.264/AVC streams, which had the highest data hiding efficiency among the existing schemes. Xiao et al. [34] proposed a reversible data hiding scheme based on multiple two-dimensional histogram modifications for JPEG images and a new coefficient pairing strategy. The scheme added a smaller file size and higher visual quality. Zhao et al. [35] avoided distortion drift caused by hidden data in videos and increased the hiding performance by selecting any embedded block in the discrete cosine transform brightness block and selecting two coefficients as hiding factor pairs. Chen et al. [36] enhanced the histogram shift scheme to achieve a high capacity based on a two-dimensional histogram to encrypt videos with a low embedded capacity.

Some of the above schemes increased the embedded capacity while decreasing the embedding rate, and some increased the video quality while increasing the file size. To solve the problem that the peak signal-to-noise ratio (PSNR) and the structural similarity index measure (SSIM) of videos are not well measured, whereas the embedded capacity is guaranteed, we present a new multihistogram shift scheme. First, the scheme transcodes the YUV file to obtain the H.264 video stream, then it decodes the DCT block by entropy, and then it adaptively finds the DCT block, which can be embedded in the data, and obtains the DCT coefficient. Secondly, the scheme divides the coefficients into the shift and embedding coefficients to generate one- and two-dimensional histograms, respectively. Then, the new two-dimensional histogram shift scheme that we propose embeds the data. This scheme can maintain a high peak signal-to-noise ratio and structural similarity index metrics while guaranteeing a certain embedded capacity, which can increase the visual quality of videos.

## 3. High-Embedded Low-Distortion Multihistogram Shift Scheme Based on DCT Coefficient

Our main aim is to classify the coefficients into the shift and embedding coefficients to generate a one- and two-dimensional histogram before generating the histogram. In this paper, a new two-dimensional histogram embedding scheme with high embedding and low distortion is proposed. Since we could generate multiple histograms, we could make the histogram clearer and thus increase their visual quality. We could also use the proposed two-dimensional histogram shift strategy to obtain an enhanced embedded capacity and peak signal-to-noise ratio.

Figure 1 shows the data embedding, data extraction, and video recovery process of this scheme. First, the scheme encoded the YUV video file to obtain the H.264 video stream, and then it obtained the DCT block with entropy decoding. It selected the DCT coefficients in the embeddable DCT blocks according to the number of zero coefficients, and it divided the coefficients into the shift and embedding coefficients to generate different histograms. It only used one-dimensional histograms for shifting and embedding secret data into two-dimensional histograms. Finally, it entropy-coded the modified block to generate a video stream with secret data. Then, it reversed the data extraction process. First, it decoded the video stream with the secret data using entropy to obtain the DCT blocks and coefficients. Then, it restored the DCT coefficients according to the inverse process of the embedded scheme, and it extracted the secret data. Finally, it obtained the H.264 video stream by entropy encoding.

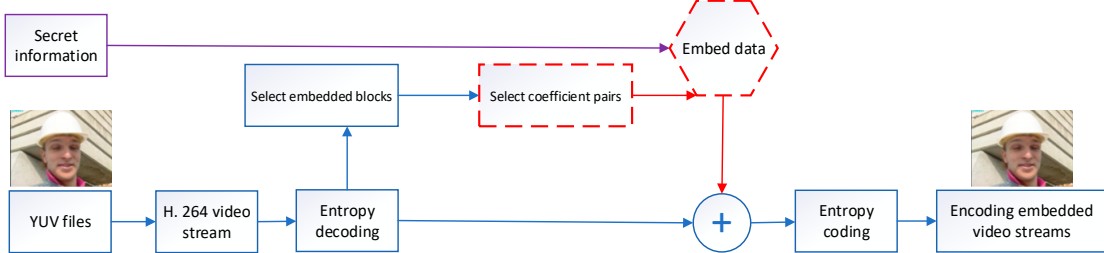

**Figure 1.** Flowchart of a reversible data hiding algorithm based on multiple histograms.

### 3.1. Constructing a Multihistogram Strategy Based on DCT Coefficients

In the previous JPEG reversible data hiding scheme [34] based on multiple two-dimensional histograms, the scheme combined each of the two adjacent nonzero AC coefficients in each DCT block to produce a coefficient pair according to the Z word scanning order. However, because many zero DCT coefficients exist in the DCT block, we propose the combination of every two adjacent coefficients in the DCT block to produce a pair of coefficients. Smaller coefficients are called embedding coefficients, and larger coefficients are called shift coefficients. Considering that the coefficients in DCT blocks are usually embedding coefficients, scholars recommend that the embedding coefficients be paired, and we used the shift coefficients to generate one-dimensional histograms for a clearer two-dimensional histogram. For each DCT block, we first classified the coefficients into an embedding coefficient set $S_i$ and shift factor set $L_i$ as follows:

$$\begin{cases} S_i = (x_i : -2 \leq x_i \leq 2 \text{ and } x_i \neq 0) \\ L_i = (x_i : x_i \geq 3 \text{ and } x_i \leq -3) \end{cases} \tag{1}$$

where $i$ represents a block index and each $i$ is constructed with two separate histograms $h_i$ and $f_i$; the coefficients in $S_i$ are paired to construct a two-dimensional histogram $h_i$; and the coefficients in set $L_i$ are simply collected to construct a one-dimensional histogram $f_i$. The details are shown in Figure 2:

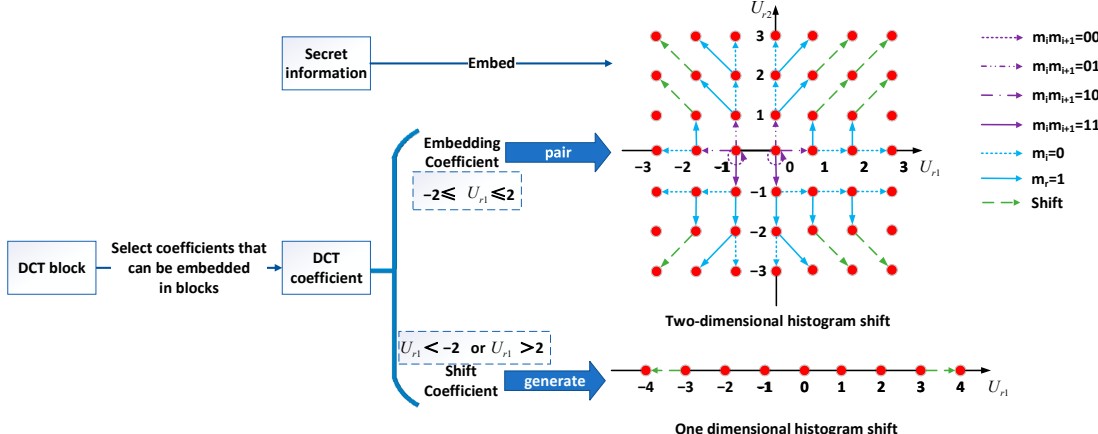

**Figure 2.** Entire histogram construction strategy diagram.

### 3.2. Multihistogram Shift Strategy Based on DCT Coefficients

Based on the proposed mapping rules, we divided the set of all the values of a DCT coefficient pair into 18 subsets because we divided the coefficients into the shift and embedding coefficients ahead of time. The shift factor produced a one-dimensional histogram that we did not use to embed the data, and the modifications were all 1, so we divided it into two subsets: greater than 2 and less than −2. We used the two-dimensional

histogram generated with the embedding factor to embed the information, and we divided it into 16 subsets for comparison with the traditional two-dimensional histogram method.

$$
\begin{aligned}
&D_1 = \{(0,0)\}, D_2 = \{(-1,0)\}, D_3 = \{(-1,-1)\}, D_4 = \{(0,-1)\}, \\
&D_5 = \{(U_{r1},0)|0 < U_{r1} < 3\}, D_6 = \{(U_{r1},0)|-3 < U_{r1} < -1\}, \\
&D_7 = \{(U_{r1},-1)|0 < U_{r1} < 3\}, D_8 = \{(U_{r1},-1)|-3 < U_{r1} < -1\}, \\
&D_9 = \{(0,U_{r2})|0 < U_{r2} < 3\}, D_{10} = \{(0,U_{r2})|-3 < U_{r2} < -1\}, \\
&D_{11} = \{(-1,U_{r2})|0 < U_{r2} < 3\}, D_{12} = \{(-1,U_{r2})|-3 < U_{r2} < -1\}, \\
&D_{13} = \{(U_{r1},U_{r2})|0 < U_{r1} < 3, 0 < U_{r2} < 3\}, \\
&D_{14} = \{(U_{r1},U_{r2})|0 < U_{r1} < 3, -3 < U_{r2} < -1\}, \\
&D_{15} = \{(U_{r1},U_{r2})|-3 < U_{r1} < -1, 0 < U_{r2} < 3\}, \\
&D_{16} = \{(U_{r1},U_{r2})|-3 < U_{r1} < -1, -3 < U_{r2} < -1\}, \\
&D_{17} = \{U_r|U_r \geq 3\}, D_{18} = \{U_r|U_r \leq -3\}.
\end{aligned}
\tag{2}
$$

where $U_r$ is the DCT coefficient, $U_{r1}$ is the DCT coefficient on the x-axis, $U_{r2}$ is the DCT coefficient on the y-axis, and D represents the set of all DCT coefficient pairs.

If the coefficient pair is $(U_{r1}, U_{r2}) \in D_1$, the marked coefficient pair $(U'_{r1}, U'_{r2})$ will be expressed as:

$$
(U'_{r1}, U'_{r2}) = \begin{cases}
(U_{r1}, U_{r2}), & \text{if } m_i m_{i+1} = 00 \\
(U_{r1}, U_{r2} + 1), & \text{if } m_i m_{i+1} = 01 \\
(U_{r1} + 1, U_{r2}), & \text{if } m_i m_{i+1} = 10 \\
(U_{r1}, U_{r2} - 1), & \text{if } m_i m_{i+1} = 11
\end{cases}
\tag{3}
$$

If the coefficient pair is $(U_{r1}, U_{r2}) \in D_2$, the marked coefficient pair $(U'_{r1}, U'_{r2})$ will be expressed as:

$$
(U'_{r1}, U'_{r2}) = \begin{cases}
(U_{r1}, U_{r2}), & \text{if } m_i m_{i+1} = 00 \\
(U_{r1}, U_{r2} + 1), & \text{if } m_i m_{i+1} = 01 \\
(U_{r1} - 1, U_{r2}), & \text{if } m_i m_{i+1} = 10 \\
(U_{r1}, U_{r2} - 1), & \text{if } m_i m_{i+1} = 11
\end{cases}
\tag{4}
$$

If the coefficient pair is $(U_{r1}, U_{r2}) \in D_3$ or $(U_{r1}, U_{r2}) \in D_8$, the marked coefficient pair $(U'_{r1}, U'_{r2})$ will be expressed as:

$$
(U'_{r1}, U'_{r2}) = \begin{cases}
(U_{r1} - 1, U_{r2}), & \text{if } m_i = 0 \\
(U_{r1}, U_{r2} - 1), & \text{if } m_i = 1
\end{cases}
\tag{5}
$$

If the coefficient pair is $(U_{r1}, U_{r2}) \in D_4$ or $(U_{r1}, U_{r2}) \in D_7$, the marked coefficient pair $(U'_{r1}, U'_{r2})$ will be expressed as:

$$
(U'_{r1}, U'_{r2}) = \begin{cases}
(U_{r1} + 1, U_{r2}), & \text{if } m_i = 0 \\
(U_{r1}, U_{r2} - 1), & \text{if } m_i = 1
\end{cases}
\tag{6}
$$

If the coefficient pair is $(U_{r1}, U_{r2}) \in D_5$, the marked coefficient pair $(U'_{r1}, U'_{r2})$ will be expressed as:

$$
(U'_{r1}, U'_{r2}) = \begin{cases}
(U_{r1} + 1, U_{r2}), & \text{if } m_i = 0 \\
(U_{r1}, U_{r2} + 1), & \text{if } m_i = 1
\end{cases}
\tag{7}
$$

If the coefficient pair is $(U_{r1}, U_{r2}) \in D_6$, the marked coefficient pair $(U'_{r1}, U'_{r2})$ will be expressed as:

$$
(U'_{r1}, U'_{r2}) = \begin{cases}
(U_{r1} - 1, U_{r2}), & \text{if } m_i = 0 \\
(U_{r1}, U_{r2} + 1), & \text{if } m_i = 1
\end{cases}
\tag{8}
$$

If the coefficient pair is $(U_{r1}, U_{r2}) \in D_9$, the marked coefficient pair $(U'_{r1}, U'_{r2})$ will be expressed as:

$$
(U'_{r1}, U'_{r2}) = \begin{cases}
(U_{r1}, U_{r2} + 1), & \text{if } m_i = 0 \\
(U_{r1} + 1, U_{r2} + 1), & \text{if } m_i = 1
\end{cases}
\tag{9}
$$

If the coefficient pair is $(U_{r1}, U_{r2}) \in D_{10}$, the marked coefficient pair $(U'_{r1}, U'_{r2})$ will be expressed as:

$$(U'_{r1}, U'_{r2}) = \begin{cases} (U_{r1}, U_{r2} - 1), & \text{if } m_i = 0 \\ (U_{r1} + 1, U_{r2} - 1), & \text{if } m_i = 1 \end{cases} \tag{10}$$

If the coefficient pair is $(U_{r1}, U_{r2}) \in D_{11}$, the marked coefficient pair $(U'_{r1}, U'_{r2})$ will be expressed as:

$$(U'_{r1}, U'_{r2}) = \begin{cases} (U_{r1}, U_{r2} + 1), & \text{if } m_i = 0 \\ (U_{r1} - 1, U_{r2} + 1), & \text{if } m_i = 1 \end{cases} \tag{11}$$

If the coefficient pair is $(U_{r1}, U_{r2}) \in D_{12}$, the marked coefficient pair $(U'_{r1}, U'_{r2})$ will be expressed as:

$$(U'_{r1}, U'_{r2}) = \begin{cases} (U_{r1}, U_{r2} - 1), & \text{if } m_i = 0 \\ (U_{r1} - 1, U_{r2} - 1), & \text{if } m_i = 1 \end{cases} \tag{12}$$

If the coefficient pair is $(U_{r1}, U_{r2}) \in (D_{13}, D_{14}, D_{15}, D_{16})$, the data cannot be embedded, and the marked coefficient pair $(U'_{r1}, U'_{r2})$ will be expressed as:

$$(U'_{r1}, U'_{r2}) = \begin{cases} (U_{r1} + 1, U_{r2} + 1), & \text{if } (U_{r1}, U_{r2}) \in D_{13} \\ (U_{r1} + 1, U_{r2} - 1), & \text{if } (U_{r1}, U_{r2}) \in D_{14} \\ (U_{r1} - 1, U_{r2} + 1), & \text{if } (U_{r1}, U_{r2}) \in D_{15} \\ (U_{r1} - 1, U_{r2} - 1), & \text{if } (U_{r1}, U_{r2}) \in D_{16} \end{cases} \tag{13}$$

If the coefficient pair is $U_r \in D_{17}$ or $U_r \in D_{18}$, the marked coefficient pair $U'_r$ will be expressed as:

$$(U'_{r1}, U'_{r2}) = \begin{cases} U_r + 1, & \text{if } U_r \in D_{17} \\ U_r - 1, & \text{if } U_r \in D_{18} \end{cases} \tag{14}$$

If the coefficient pair $(U_{r1}, U_{r2})$ belongs to the $D_1$ to $D_{12}$ set, Figure 3 shows the DCT coefficient shift scheme that we propose.

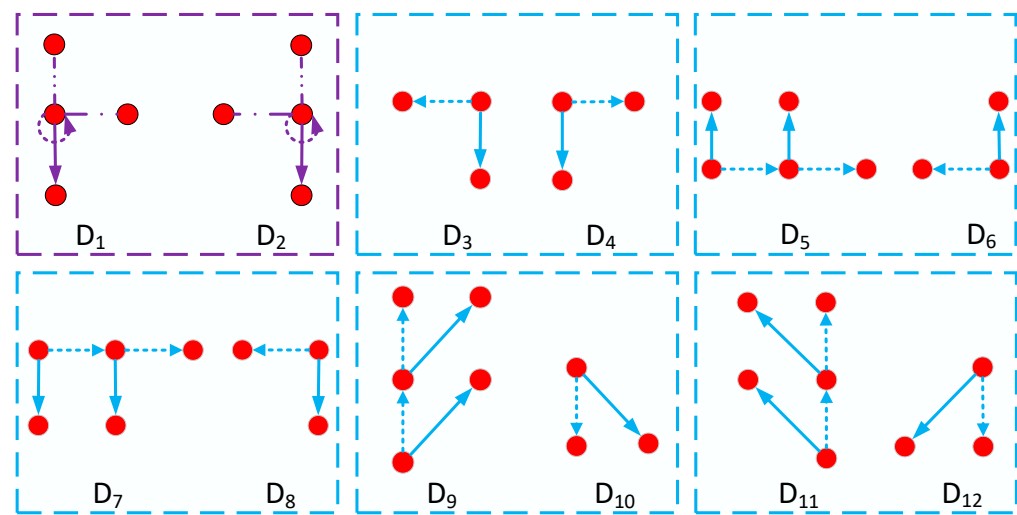

**Figure 3.** Coefficient shift strategy for $D_1 \sim D_{12}$ set.

If the coefficient pairs $(U_{r1}, U_{r2})$ belong to $D_{13}, D_{14}, D_{15}, D_{16}, D_{17}, D_{18}$, we only used the coefficients of these sets in this scheme for shifting, not for embedding the data.

Based on the two-dimensional histogram shift strategy that we propose, we embedded the data in the set and extracted them from the new set. The embedded capacity of the multiple histogram shift can be calculated with Equation (15):

$$EC_{pre} = 2 \sum_{(U_{r1}, U_{r2}) \in D_1 \cup D_2} t(U_{r1}, U_{r2}) + \sum_{(U_{r1}, U_{r2}) \in D_3 \cup D_4 \cup D_5 \cup D_6 \cup D_7 \cup D_8 \cup D_9 \cup D_{10} \cup D_{11} \cup D_{12}} t(U_{r1}, U_{r2}) \tag{15}$$

$EC_{pre}$ is the embedding capacity of this scheme. Based on the traditional two-dimensional histogram embedding capacity and Equation (15), the embedded capacity difference between the proposed multihistogram shift and the traditional two-dimensional histogram shift can be calculated:

$$EC_{con} - EC_{pre} = \sum_{(U_{r1}, U_{r2}) \in D_3 \cup D_4} t(U_{r1}, U_{r2}) \tag{16}$$

$EC_{con}$ is the embedding capacity of traditional solutions. The embedding distortion of the multihistogram shift can be calculated with Equation (17):

$$\begin{aligned} ED_{pre} = \tfrac{3}{4} \sum_{(U_{r1}, U_{r2}) \in D_1 \cup D_2} t(U_{r1}, U_{r2}) &+ \sum_{(U_{r1}, U_{r2}) \in D_3 \cup D_4 D_5 \cup D_6 \cup D_7 \cup D_8} t(U_{r1}, U_{r2}) \\ + \tfrac{3}{2} \sum_{(U_{r1}, U_{r2}) \in D_9 \cup D_{10} \cup D_{11} \cup D_{12}} t(U_{r1}, U_{r2}) &+ 2 \sum_{(U_{r1}, U_{r2}) \in D_3 \cup D_{14} \cup D_{15} \cup D_{16}} t(U_{r1}, U_{r2}) \\ + \sum_{U_r \in D_{17} \cup D_{18}} t(U_r) & \end{aligned} \tag{17}$$

$ED_{pre}$ is the embedding distortion of this scheme. Based on the traditional two-dimensional histogram embedding distortion and Equation (17), the hidden distortion difference between the proposed multiple and traditional histogram shift can be calculated:

$$\begin{aligned} ED_{con} - ED_{pre} = \tfrac{1}{4} \sum_{(U_{r1}, U_{r2}) \in D_1 \cup D_2} t(U_{r1}, U_{r2}) &+ \tfrac{1}{2} \sum_{(U_{r1}, U_{r2}) \in D_5 \cup D_6 \cup D_7 \cup D_8} t(U_{r1}, U_{r2}) \\ + \sum_{U_r \in D_{17} \cup D_{18}} t(U_r) & \end{aligned} \tag{18}$$

$E_{con}$ is the embedding distortion of traditional solutions. Based on Equations (16) and (18), one can see that this scheme had a lower embedded distortion than the traditional scheme, and we could roughly estimate that this scheme had a higher PSNR and SSIM. When the values of the coefficient pairs ($U_{r1}$, $U_{r2}$) belong to subsets $D_1$, $D_2$, $D_5$, $D_6$, $D_7$, and $D_8$, both the traditional and present schemes can obtain the same embedded capacity, but this scheme can result in at most one modification and can obtain a lower embedded distortion than the traditional scheme with fewer modifications. When $U_r$ is a subset of $D_{17}$ and $D_{18}$, it is only used for shifting, and very few shift coefficients are present in DCT blocks, which can be ignored or forgotten. Moreover, this scheme results in at most one modification in these two subsets, which reduces distortion and increases efficiency.

### 3.3. Data Embedding

This scheme embeds secret data into a pair of coefficients based on a multiple histogram shift, as shown in Figure 1. The main process of data embedding includes the following five steps:

(1)  Use the encoder to obtain an H.264 video stream from a YUV video file, and use entropy decoding to obtain the DCT block;

(2)  Select the DCT coefficients in the embeddable DCT blocks and divide the coefficients into shift (greater than 2 or less than −2) and embedding coefficients (greater than or equal to −2 and less than or equal to 2);

(3)  Generate one- and two-dimensional histograms from the shift and embedding factors, respectively. Use one-dimensional histograms to shift only to achieve reversibility and create space, and use two-dimensional histograms to embed secret data;

(4)  Embed the secret data into the two-dimensional histogram using the two-dimensional histogram embedding scheme that we propose;

(5)  Use the entropy encoding of the videos embedded in secret data to obtain video streams with secret data.

### 3.4. Data Extraction and Video Recovery

Figure 1 shows the data extraction and video recovery process of this scheme, which consists of three main steps:

(1) Entropy decode the video streams with secret data to obtain DCT blocks;
(2) Extract the data to obtain the secret data according to the reverse embedding process;
(3) After data extraction, encode the video with entropy to obtain the H.264 video stream, and then convert the H.264 video stream to a YUV video file using the encoder.

## 4. Experimental Results and Analysis

To evaluate the performance of the proposed algorithm, we implemented all the experiments on a Windows 1064-bit operating system using Intel(R)Core(TM)i7-6700HQ with 8 GB of RAM and applied them to the H.264/AVC reference software JM12.2. We used seven standard video sequences, namely Carphone, Foreman, Container, News, Crew, Hall, and Mobile, with a resolution of $176 \times 144$ and a frame rate of 30 frames per second. IBPBPBPBPB was the picture group. In addition to the above, the other encoding parameters retained their default values.

### 4.1. Embedded Capacity

Figure 4 shows the distribution of Foreman's DCT coefficients with QP = 28 using this method. Almost all the values of the DCT coefficients were between −1, 0, and 1, and most of them were around 0. Since this scheme mainly uses embedding coefficients to embed the data, embedding data with this scheme can result in a higher amount of embedding when many embedding coefficients are present.

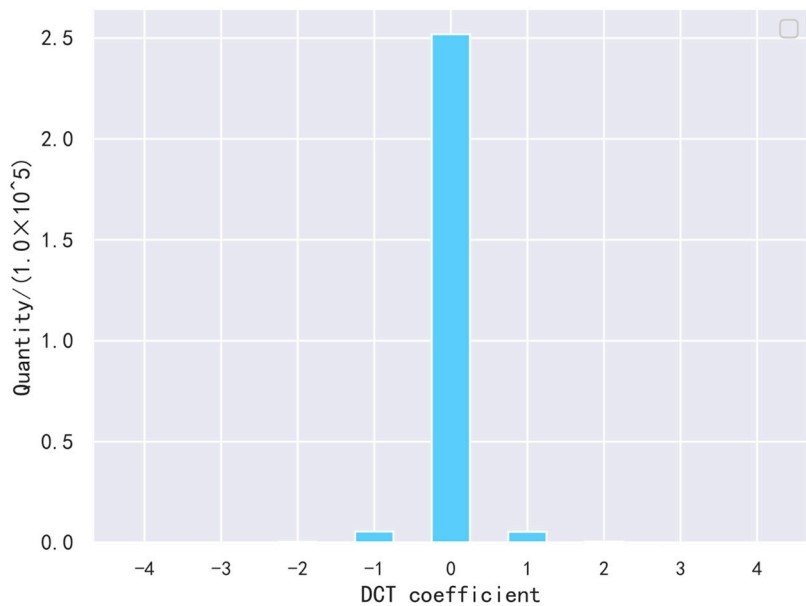

**Figure 4.** Distribution of DCT coefficient values.

Table 1 shows the maximum embedding capacity for each video encoding with a quantization parameter (QP) value of 28. As shown in the table, each video had different embedding capabilities, which may depend to a large extent on the video content because different numbers of coefficients used for data embedding were present in each video, resulting in different embedding capacities. As shown in Table 1, this method had a larger embedded capacity in Crew and Mobile videos than in other videos. Since Mobile and Crew videos are relatively intense and have more complex textures, they have more coefficients for data embedding than other video sequences. As shown in the table, this scheme had a larger embedding capacity than the control scheme. Compared with the scheme in [29], the average embedded capacity increased by 10,456 bits, and the percentage increased by

77.60%. Compared with the scheme in [37], the percentage increased by 17.26%, and the average embedded capacity increased by 3522 bits.

**Table 1.** Comparison of the embedded capacity of seven video sequences with other schemes.

| Video Sequence | Embedded Capacity/Bit | | |
|---|---|---|---|
| | Chen [29] | Xu [37] | Proposed |
| Carphone | 10,843 | 9264 | 11,652 |
| Foreman | 9705 | 8061 | 11,047 |
| Container | 9807 | 8762 | 10,956 |
| News | 8386 | 7231 | 8183 |
| Crew | 21,822 | 47,651 | 54,972 |
| Hall | 8394 | 7356 | 8213 |
| Mobile | 25,363 | 54,531 | 62,486 |
| Average | 13,474 | 20,408 | 23,930 |

### *4.2. Visual Quality*

To assess the impact of the hidden data on video quality, we must consider two important aspects: subjective visual perception and objective indicators. The objective indicators that scholars use to assess the perceived quality are mainly the peak signal-to-noise ratio (PSNR) and structural similarity index measure (SSIM).

### 4.2.1. Subjective Visual Perception

Since the embedding scheme that we used is reversible, the original video can be restored without distortion after data extraction. We randomly selected one frame from the seven video sequences without embedded data for observation, as shown in Figure 5. Then, we selected the corresponding marker frame from the video sequence after data embedding, as shown in Figure 6. The original video frame and the marked video frame could not be distinguished from the subjective visual experience. The visual quality of the other frames was similar, so we did not display the visual effects of all the frames here.

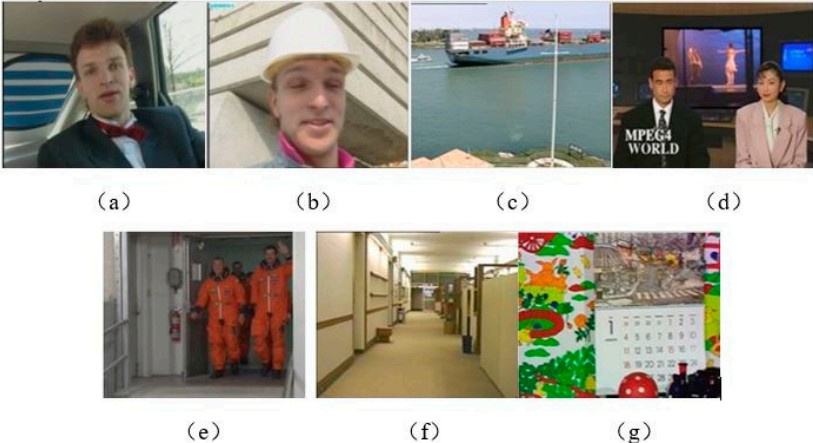

**Figure 5.** Original video frame: (**a**) Carphone, (**b**) Foreman, (**c**) Container, (**d**) News, (**e**) Crew, (**f**) Hall, and (**g**) Mobile.

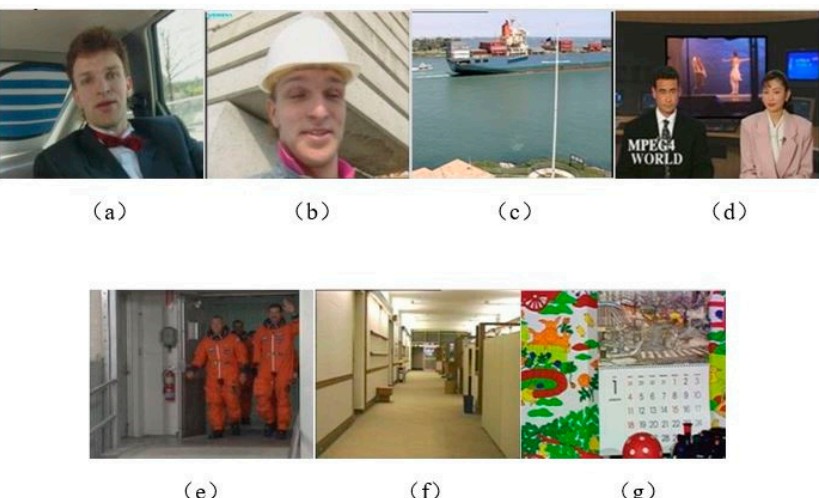

**Figure 6.** Video frame after data hiding: (**a**) Carphone, (**b**) Foreman, (**c**) Container, (**d**) News, (**e**) Crew, (**f**) Hall, and (**g**) Mobile.

### 4.2.2. Structural Similarity Index Measure (SSIM)

The structural similarity index measure is an objective indicator that scholars use to measure visual quality. The SSIM value is between zero and one. The closer it is to zero, the more different the reference video is from the target, and the closer it is to one, the more similar the reference video and target are.

Table 2 shows the SSIM values of the original video sequence, the other three schemes, and this scheme when the QP value is 28. As shown in the table, no remarkable fluctuation was present between this scheme and the original video sequence, which means that the video quality had not decreased according to the structural similarity index.

**Table 2.** Comparison of SSIM values of seven video sequences with other schemes.

| Video Sequence | SSIM | | | | |
|:---:|:---:|:---:|:---:|:---:|:---:|
| | Original | Chen [27] | Xu [32] | Li [38] | Proposed |
| Carphone | 0.97 | 0.97 | 0.97 | 0.97 | 0.97 |
| Foreman | 0.96 | 0.96 | 0.95 | 0.95 | 0.96 |
| Container | 0.93 | 0.93 | 0.93 | 0.93 | 0.93 |
| News | 0.97 | 0.97 | 0.97 | 0.97 | 0.97 |
| Crew | 0.95 | 0.95 | 0.95 | 0.95 | 0.95 |
| Hall | 0.97 | 0.97 | 0.96 | 0.96 | 0.97 |
| Mobile | 0.97 | 0.97 | 0.97 | 0.97 | 0.97 |
| Average | 0.96 | 0.96 | 0.96 | 0.96 | 0.96 |

### 4.2.3. Peak Signal-to-Noise Ratio (PSNR)

Secondly, the PSNR is also an important indicator for evaluating visual perception quality. The larger the PSNR, the higher the quality. When the PSNR is higher than 40 dB, the produced image is very close to the original image. When the PSNR is 30–40 dB, distortion is present but acceptable. When the PSNR is below 30, the quality is very poor and unacceptable.

As shown in Table 3, when the QP is 28, the original PSNR value is greater than the PSNR of the three control schemes and this scheme. According to the data in Table 3, the average PSNR values of Chen [27], Xu [32], Li [38], and this scheme were 34.46, 36.22, 36.15, and 37.21 dB, respectively. Thus, the PSNR value of this scheme was larger, and we

obtained higher visual quality. Additionally, according to Tables 1 and 2, the scheme that we propose had a higher embedding capacity when the SSIM was essentially the same.

**Table 3.** PSNR values of seven video sequences compared with other schemes.

| Video Sequence | PSNR | | | | |
| --- | --- | --- | --- | --- | --- |
| | Original | Chen [27] | Xu [32] | Li [38] | Proposed |
| Carphone | 38.74 | 35.67 | 37.42 | 37.39 | 38.38 |
| Foreman | 37.31 | 32.83 | 36.37 | 36.28 | 37.07 |
| Container | 37.42 | 33.65 | 36.49 | 36.45 | 37.11 |
| News | 38.46 | 35.23 | 37.46 | 37.39 | 38.24 |
| Crew | 37.87 | 35.51 | 36.37 | 36.32 | 37.19 |
| Hall | 38.50 | 36.16 | 37.37 | 37.31 | 38.23 |
| Mobile | 34.85 | 32.20 | 32.09 | 31.93 | 34.26 |
| Average | 37.59 | 34.46 | 36.22 | 36.15 | 37.21 |

In addition, we randomly selected Container, Foreman, and Mobile standard videos to compare the PSNR performance of fixed capacity. The results are shown in Figures 7–9. In the four RDH schemes, when the data with the same capacity were hidden in the video sequence, the scheme could obtain a larger PSNR value. By essentially combining the same SSIM values in Table 2, one can conclude that the scheme we propose can obtain the best video quality with the same embedding amount. Figures 8 and 9 show that the PSNR value obtained for the Mobile video sequence was small, whereas the PSNR value obtained for the Foreman video sequence was large. Since the Mobile video had more color elements and a greater texture complexity than the Foreman video, the PSNR value was smaller. Thus, the more color elements and complex textures present in the video, the smaller the PSNR value obtained and the worse the video quality.

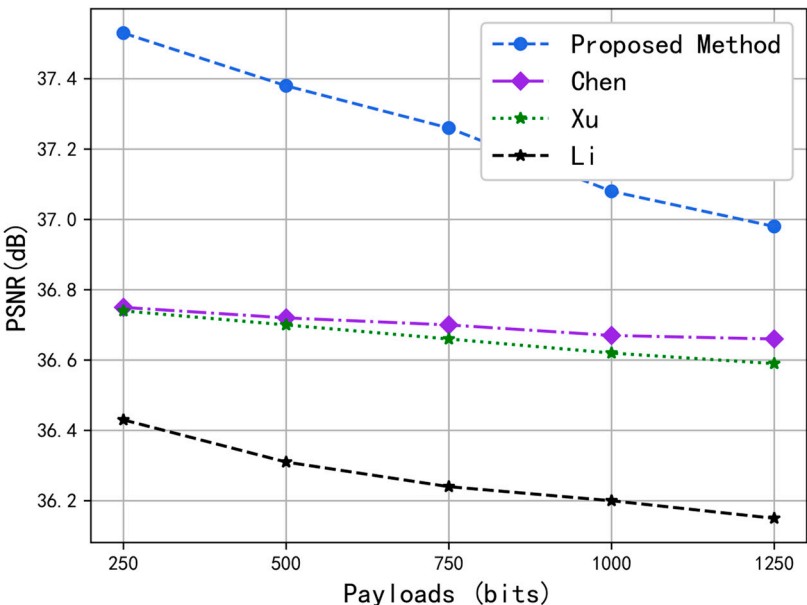

**Figure 7.** Comparison of PSNR values of different schemes on Container.

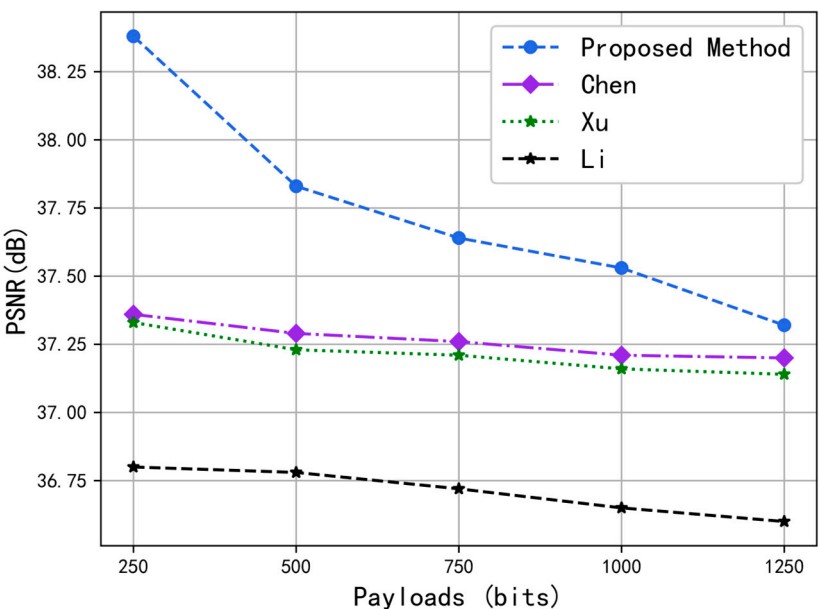

**Figure 8.** Comparison of PSNR values of different schemes on Foreman.

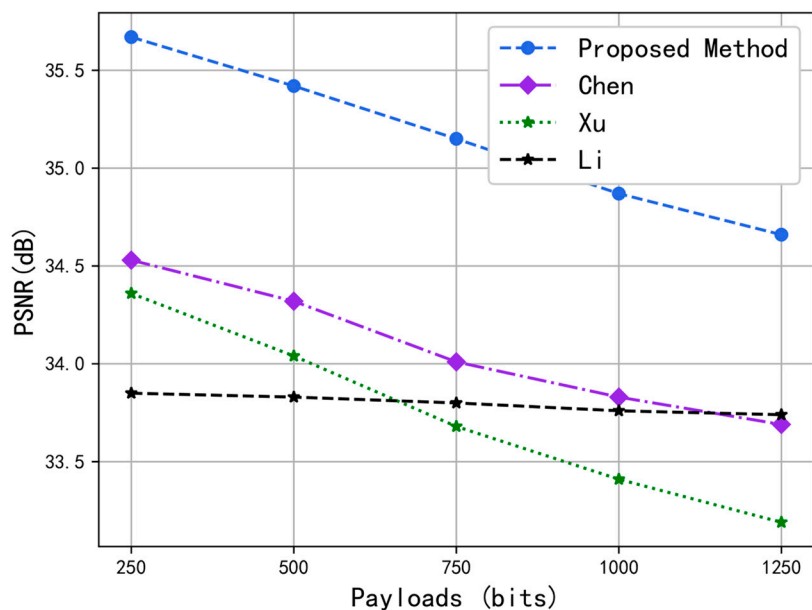

**Figure 9.** Comparison of PSNR values of different schemes on Mobile.

### 5. Conclusions

In this paper, a reversible data hiding scheme for H.264/AVC videos based on DCT coefficients with a high-embedding and low-distortion multiple histogram shift is proposed. Based on the new construction strategy of the one- and two-dimensional histograms that we propose, the scheme can generate more clear histograms for enhanced visual perception. In this paper, a high-embedding and low-distortion two-dimensional histogram shift scheme based on DCT coefficients is proposed, which can result in enhanced embedding capacity and peak signal-to-noise ratio performance. We tested this scheme on standard video sequences and compared it with existing schemes. The experimental results showed that the embedded capacity, SSIM, and PSNR performance can be increased by using this scheme.

**Author Contributions:** Conceptualization, Y.Y. and X.X.; methodology, Y.Y.; experiment, Y.Y. and Y.L.; writing—original draft preparation, Y.Y.; writing—review and editing, Y.T. and Z.W.; funding acquisition, J.Q. and Y.T. All authors have read and agreed to the published version of the manuscript.

**Funding:** This work was supported in part by the Natural Science Foundation of Hunan Province under Grant 2022JJ31019, in part by the National Natural Science Foundation of China under Grant 62002392, and in part by the Science Research Projects of Hunan Provincial Education Department under Grant 22A0195.

**Data Availability Statement:** Not applicable.

**Conflicts of Interest:** The authors declare no conflict of interest.

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
