# Peer review of "High-Embedded Low-Distortion Multihistogram Shift Video Reversible Data Hiding Based on DCT Coefficient"

_electronics, doi:10.3390/electronics12071652_

Round 1

Reviewer 1 Report

1. How about the computational time with the proposed approach and compare methods. 

2. Table 1 shows the maximum embedding capacity, its very worthwhile to analyze. But how about the embedding capacity for video encoding with the larger value of the quantization parameter (QP).

3. Please add the reference number of comparing methods in Tables 1, 2, and 3.

Author Response

请参阅附件。

Reviewer 2 Report

1) Authors should provide a summary of results outcomes in a table with the related works to show improvement.

2)Some of the references in this paper are outdated, some of them are even more than 10 years from now. The authors should update references with the last five years papers.

3) What the variables and/or constants in the related equation represent should be stated after the equation.

4) Revise the size and spaces of equations and the defined parameters.

5) Fig 4, 7, 8 and 9 are not clear; authors should redraw it with high resolution.

6) How multihistogram shift scheme based on DCT coefficients works? Elaborate in detail.

7) In line number 219, what D1 to D18 are indicates. Explain.

Reviewer 3 Report

The authors proposed a High-Embedded Low-Distortion Multihistogram Shift Video Reversible Data Hiding Based on DCT Coefficient. It is an interesting work but I have the following concerns,

1. What are the motivations behind the proposed system / what are the drawbacks of existing systems?

2. The related work section is very poor and is not mentioning the most relevant recent research works.

3. How the anti-aliasing problem of DCT is overcome in your proposed system?

4. Is embedding multiple watermarks in your system possible? Give your comment on this. Refer to the popular multiple watermarks systems to address this query (An Adaptive HVS Based Video Watermarking scheme for Multiple Watermarks using BAM Neural Networks and Fuzzy Inference System, A Bi-directional Associative Memory based Multiple Image Watermarking on Cover Video).

5. The experimental section is poor and needs robustness to be included. Any Reversible hiding system has to test its robustness using NCC and BER and it is missed in the proposed model. Kindly refer to the articles that might help to address this query (http://link.springer.com/article/10.1007/s11042-015-2789-9, https://doi.org/10.1007/s11042-018-5800-4)

6. Finally, I can find a lot of grammatical and spelling errors and needs your attention in that.

Overall, it will be good work if it addresses all these queries. 

Reviewer 4 Report

The authors present a reversible data hiding scheme for H.264/AVC videos based on DCT coefficients with a high-embedding and low-distortion multiple histogram shift. To solve the problem that the peak signal-to-noise ratio (PSNR) and the structural similarity index measurement (SSIM) of videos are not well measured whereas the embedded capacity is guaranteed, the authors present a new multi-histogram shift scheme. First, the scheme transcodes the YUV file to obtain the H.264 video stream, then it decodes the DCT block by entropy, and then it adaptively finds the DCT block, which can be embedded in the data, and obtains the DCT coefficient. Secondly, the scheme divides the coefficients into shift and embedding coefficients to generate one- and two-dimensional histograms, respectively. Then, the new two-dimensional histogram shift scheme that they propose embeds the data. This scheme can maintain a high peak signal-to-noise ratio and structural similarity index metrics while guaranteeing a certain embedded capacity, which can increase the visual quality of videos. 

My concern is about developing the same scheme for HEVC and VVC (266)! is it possible to use your scheme with the FNST transfom used in VVC? 

Round 2

Reviewer 2 Report

All the corrections and comments are addressed satisfactorily.

Reviewer 3 Report

The article has improved a lot in this round of revision and authors have responded to most of my queries. But I have minor comments,

1. Avoid using the pronoun "We" under contributions and correct it.

2. Section 2 should be "Related Works"

3. The resolution of figure 6. need to be improved.

4. What is the future scope of this research?

5. Still, I can find some spelling errors. For example, " impeccability" in 9th line of Introduction. " [26] hid" in page 3

Author Response

请参阅附件。
